# Multimodal Analgesia for Accelerated Rehabilitation after Total Knee Arthroplasty: A Randomized, Double-Blind, Controlled Trial on the Effect of the Co-Application of Local Infiltration Analgesia and Femoral Nerve Block Combined with Dexmedetomidine

**DOI:** 10.3390/brainsci12121652

**Published:** 2022-12-02

**Authors:** Chen Gao, Tingting Huang, Kerong Wu, Wei Zhang, Sheng Wang, Xiaoqing Chai, Yanhu Xie, Chaoliang Tang

**Affiliations:** 1Department of Anesthesiology, The First Affiliated Hospital of USTC, Division of Life, Sciences and Medicine, University of Science and Technology of China, Hefei 230001, China; 2Department of Anesthesiology, The First Affiliated Hospital of Anhui Medical University, Hefei 230001, China; 3Department of Orthopedics, The First Affiliated Hospital of USTC, Division of Life, Sciences and Medicine, University of Science and Technology of China, Hefei 230001, China

**Keywords:** total knee arthroplasty, local infiltration analgesia, femoral nerve block, dexmedetomidine

## Abstract

Background: Multimodal postoperative pain regimens are widely used following total knee arthroplasty (TKA). However, there are few studies on the rehabilitation of the co-application of local infiltration analgesia (LIA) and femoral nerve block (FNB) combined with dexmedetomidine (DEX) for patients undergoing TKA. This study aimed to investigate the effect of LIA plus FNB and co-application of perioperative DEX on TKA outcomes. Methods: 95 patients were randomized into two groups. Patients in group B (*n* = 48) received a single preoperative FNB and LIA. Patients in group A (*n* = 47) received FNB and LIA, as well as continuous intravenous injection of DEX starting from the induction of anesthesia to postoperative day 2. All patients were allowed patient-controlled analgesia postoperatively. Visual analog scale (VAS) scores, knee range of motion (ROM) degrees, narcotic consumption, length of hospital stay (LOS), complications, Hospital for Special Surgery (HSS) scores and Montreal Cognitive Assessment-Basic (MoCA-B) Scores were recorded. Results: In group A, the mean VAS scores at rest and during movement were lower, the amount of rescue analgesia was decreased, first time of ambulation was reduced, ROM was improved, MoCA-B Scores were increased, LOS was shorter, HSS scores were higher postoperatively compared with group B (all *p* < 0.05). Conclusion: Our study indicated multimodal analgesia involving a single FNB and LIA combined with DEX accelerates rehabilitation for patients undergoing TKA.

## 1. Introduction

Osteoarthritis (OA) is a degenerative change of articular cartilage generally characterized by pain, deformity, instability and functional limitation. It usually affects elderly patients and is the main cause of adult disability around the world [1]. According to the World Health Organization (WHO), more than 343 million people are affected by some form of osteoarthritis, and the incidence rate of women is higher than that of men [1,2]. Patient education (PE), exercise therapy and weight management are recommended as first-line interventions for hip and knee osteoarthritis (OA) [3]. PE can involve some form of communication, including improving patient knowledge, developing self-management and life skills, which are conducive to the health of individuals and communities [3]. PE can effectively reduce pain and improve function of the arthrosis [3]. Researches showed that combining conservative treatment such as exercise therapy or manual therapy with PE should be encouraged, taking into account the statistical advantages and clinical importance to improve the prognosis of patients [3,4].

Total knee arthroplasty (TKA) is a highly effective way to treat end-stage and severe arthritis [5,6]. It improves the function of the knee joint and quality of life. However, TKA is also a traumatic procedure that can lead to severe postoperative pain, activity limitations, and decrease in the effectiveness of functional exercise [7,8,9]. Inadequate pain control seriously affects postoperative rehabilitation, increases the incidence of complications, extends the length of hospital stay (LOS), and increases the cost of hospitalization [9,10]. Moreover, insufficient analgesia aggravates postoperative cognitive dysfunction (POCD) [11,12]. POCD is associated not only with anesthetic and types of surgery but also immediate postoperative pain. Pain management strategy is a significant risk factor for POCD, and pain-alleviating approaches would enhance postoperative cognitive performance in patients [13]. 

Numerous multimodal analgesia regimens are available that can involve local infiltration analgesia (LIA), femoral nerve block (FNB), and sciatic nerve block (SNB), which are used as adjuncts to analgesic medication [8,9]. Previous studies [14,15] have demonstrated that a single FNB provides effective analgesia, decreases opioid consumption, and facilitates recovery. SNB helps to ease the pain and accelerates rehabilitation. However, adding an SNB to an FNB, especially a continuous block, might damage motor function and thereby increases the risk of falls [16,17]. Chinachoti [18] noted that adding periarticular LIA to FNB significantly enhanced postoperative pain control and patient satisfaction, and diminished narcotic consumption. LIA is considered to be an alternative to peripheral nerve block in order to reduce the risk of falls during TKA rehabilitation. 

Dexmedetomidine is a potent and highly selective transmembrane G-protein-coupled central α_2_-receptor agonist that reduces sympathetic outflow from the central nervous system and provides sedation and analgesia without inhibiting respiration [19]. Many studies have shown that dexmedetomidine (DEX) can significantly alleviate pain, extend the duration of analgesic effects, reduce opioid demand and opioid-related adverse events [20,21,22]. The incidence of postoperative nausea and vomiting (PONV) was reduced with DEX treatment [23]. In addition, DEX administered intraoperatively decreased cognitive deterioration postoperatively in elderly individuals undergoing planned laparotomy [24]. Mei [25] found that for older patients undergoing total knee or hip arthroplasty, intraoperative sedation with DEX improved neurocognitive function after surgery. However, the evidence for the adjuvant analgesic effects of DEX perioperatively combined with LIA and FNB after total knee arthroplasty is debatable. This study aimed to investigate the effects under which the combination of DEX and LIA plus FNB would alleviate pain, enhance the knee range of motion (ROM), reduce the incidence of PONV and POCD, and facilitate rapid rehabilitation for patients undergoing TKA. 

## 2. Materials and Methods

This randomized, double-blind controlled trial was approved by the Biomedical Research Ethics Committee of Anhui Medical University (No.20160177) and was registered at Chictr.org (registration No. ChiCTR-INR-16008288) prior to patient enrollment and all the subjects provided written informed consent. The trial was conducted between April 2016 and September 2020 and involved recruiting 95 patients (range: 60–80 years) who were scheduled to undergo unilateral TKA in The First Affiliated Hospital of USTC (Anhui Provincial Hospital) (Figure 1). 

### 2.1. Patients and Study Design

The inclusion criteria were as follows: patients with American Society of Anesthesiologists (ASA) physical status class I-III, scheduled for unilateral TKA, aged 60–80 years, and willing to undergo FNB and periarticular LIA. The exclusion criteria were as follows: patients with POCD preoperatively, allergic to anesthetics, with history of narcotic abuse, with infection around the puncture sites, with education level higher than junior school, with ipsilateral neuromuscular diseases/injuries, and knee revision. 

In total, 100 patients were recruited. Thus, data from 47 and 48 patients in groups A and B, respectively, were analyzed (Figure 1). The patients were randomized to one of the two groups using a computer-generated randomization schedule. Patients in group A (*n* = 47) received a single preoperative FNB and LIA after placing the knee prosthesis and received continuous infusion of DEX beginning from the induction of anesthesia with an initial dose of 1 μg/kg over 15 min, and was then administered continuously at a dose of 0.4 μg·kg^−1^·h^−1^ to about 30 min before the end of the procedure. When the maintenance infusion of DEX was stopped, the PCIA pump started to be used for 48 h postoperatively. Patients in group B (*n* = 48) received a single FNB and LIA and the same volume of normal saline at the same infusion rate during the surgery and received equal time of PCIA. 

### 2.2. Protocol

#### 2.2.1. Femoral Nerve Block

Upon arrival, in the preparation room, each patient received basic vital sign monitoring: automatic non-invasive blood pressure measurement, electrocardiogram, and pulse oximetry. Each patient received standard preoperative medication 30 min before surgery: midazolam 1 mg (iv) and parecoxib 40 mg (iv). Patients in groups A and B (laying in a supine position) received a single-injection FNB with ropivacaine using an ultrasound unit (Ultrasound System, Sonosite S-Nerve, Bothell, D.C, USA) combined with neurostimulation (Stimuplex NSH 11, B. Braun Melsungen AG, Melsungen, Germany). 

After sterile preparation, a 22 G atraumatic insulated needle (Stimuplex D, B. Braun Melsungen AG, Melsungen, Germany) was inserted at 1.5–2 cm distal to the inguinal ligament and 1.5–2.0 cm lateral to the femoral artery. A stimulation intensity of 1.0 mA at a rate of 2 Hz was used. The needle, which was inserted using the in-plane approach under ultrasound guidance, was advanced at a 45° angle to the skin until quadriceps muscle twitches were elicited (cephalad patellar movement). A bolus of 5 mL ropivacaine 0.25% was administered as a test dose, and 10 min later, the anesthetic effect was assessed by pinpricking the quadriceps area. Subsequently, a main dose of 15 mL ropivacaine 0.25% was administered after aspiration. All the procedures were performed by the same senior anesthesiologists. All patients were operated under general anesthesia with the induction of etomidate (0.3 mg/kg), sufentanil (0.4 μg/kg) and rocuronium (0.6 mg/kg). Anesthesia was maintained with propofol, remifentanil and sevoflurane.

#### 2.2.2. Local Infiltration Analgesia

During the procedure, total volume of 50 mL LIA solution including ropivacaine 150 mg, morphine 5 mg, and epinephrine 300 µg was infiltrated. Before prosthesis implantation, 20–30 mL mixture was injected into the posterior capsule and the rest of the solution was injected around the fascia, the synovium, the ligament, and the subcutaneous tissue before closure of the incision for all patients. 

#### 2.2.3. Postoperative Analgesia

For each patient, patient-controlled intravenous analgesia (PCIA) was used as rescue analgesia. PCIA pump in group A contained 2 μg/kg sufentanil, 2 μg/kg DEX and 4 mg tropisetron. The analgesia pump in group B was filled with 2 μg/kg sufentanil and 4 mg tropisetron. Each patient received a total volume of 150 mL PCIA pump with a basal infusion rate of 2 mL/h, a bolus dose of 2 mL and a lockout interval 15 min in the first 48 h postoperatively. Each patient received oral celecoxib 200 mg bid as pain medication and oral tramadol for breakthrough pain management during hospitalization, beginning on the morning of postoperative day (POD) 1. The amounts of narcotics (i.e., celecoxib, or tramadol) administered via any route on PODs 1 and 2 were converted to morphine equivalent (mg) [26]) and recorded for each patient.

#### 2.2.4. Recovery and Data Collection

Patient education was obtained and all patients were encouraged to perform therapeutic exercises including quadricep strengthening exercises and active range of motion (ROM) exercises (within the range of tolerance) after they returned to the ward. They were allowed to walk using a walker as soon as they were able. A physiotherapist blinded to group allocation recorded self-reported postoperative visual analog scale (VAS) scores. The patients rated postoperative pain with movement on PODs 1–2, at discharge, at 1 month and 1 year after surgery. Active knee ROM and Montreal Cognitive Assessment-Basic (MoCA-B) scores were measured preoperatively and on PODs 1–2, at discharge, at 1 month and 1 year post operation. Analgesic consumption was recorded for the first 2 days postoperatively. PONV and mean LOS during surgery were recorded. The patients were asked to complete a questionnaire to assess Hospital for Special Surgery (HSS) scores [27], which evaluates knee function, at 1 month and 1 year after discharge. HSS scores >85, 70–84, 60–69, and <59 represented excellent, good, moderate, and poor knee function, respectively [28].

### 2.3. Statistical Analysis

The primary statistical analysis involved the VAS scores on POD 1. We calculated that the sample size should be 38 patients per group based on an expected clinically relevant between-group difference in VAS scores of 1–2 points, a type I error rate of 0.05, and power of 90%. Based on an estimated 20% dropout risk, we recruited 100 subjects for the study. 

Numeric data were assessed for normality of distribution and equal variance and presented as means and standard deviations (SDs). Unpaired *t*-test involved independent samples for a difference in mean for continuous values: age, height, weight, BMI, remifentanil consumption, duration of surgery, narcotics consumption, first time of ambulation, VAS scores, LOS, HSS scores, MoCA-B scores and ROM. Chi-square tests of association were used to examine gender and PONV. SPSS 19.0 software (IBM, Armonk, NY, USA) was used for the statistical analysis. Two-sided *p*-values <0.05 were considered statistically significant.

## 3. Results

No between-group differences were found with regard to age, gender, body mass index, ASA classification, remifentanil consumption and surgery duration (Table 1).

### 3.1. Primary Endpoints

#### Pain Relief

There were significant differences in mean VAS pain scores at rest on POD2 (1.49 ± 0.78 vs. 1.81 ± 0.82, *p* = 0.02) and with movement during recovery exercise within the range of tolerance between the two groups on PODs 1–2 and at discharge (3.15 ± 0.59 vs. 4.56 ± 0.90, *p* = 0.00; 3.62 ± 0.64 vs. 4.81 ± 0.89, *p* = 0.00; 3.53 ± 0.88 vs. 4.13 ± 0.70, *p* = 0.00, respectively). However, there were no significant differences at 1 month and 1 year after surgery between the groups whether at rest or with movement (Table 2 and Table 3).

### 3.2. Secondary Endpoints

#### 3.2.1. Narcotic Consumption

Sufentanil and remifentanil were used intraoperatively and no significant differences were observed between the two groups (Table 1). There were significant differences in narcotic consumption (based on equivalent morphine use) between the groups on PODs 1 and 2. Patients in group A required significantly less rescue analgesia postoperatively than patients in group B (Table 4).

#### 3.2.2. First Time of Ambulation

All TKA patients postoperatively were encouraged to ambulate with the help of a walker. More patients ambulated earlier in group A in comparison with group B after the surgery. (*p* = 0.00) (Table 4).

#### 3.2.3. Length of Hospital Stay

LOS was significantly shortened in group A compared to group B (*p* = 0.000) (Table 4).

#### 3.2.4. Complications

Two main complications occurred after TKA: nausea and vomiting. There were 3 and 11 patients with postoperative nausea and vomiting (PONV) in groups A and B, respectively, and the difference was significantly higher in group B (*p* = 0.023) (Table 4). No patients were re-admitted to the hospital due to complications.

#### 3.2.5. Range of Motion

There were no preoperative differences in knee ROM regarding flexion and extension. However, the groups differed in ROM starting from POD 1 to 1 year after discharge. Patients in group A had significantly better ROM than the patients in group B (all *p* < 0.05) (Table 5).

#### 3.2.6. MoCA-B Scores

Patients in both groups showed no differences in MoCA-B scores preoperatively, Group A had a significant higher MoCA-B scores from POD1 to discharge. However, no significant differences were observed in MoCA-B scores after discharge (Table 6). 

#### 3.2.7. Hospital for Special Surgery Score

The HSS score, which is used to evaluate the clinical outcomes of TKA, showed a significant difference between two groups at 1 month (*p* = 0.00) and 1 year post-operation (*p* = 0.00) (Table 7).

## 4. Discussion

Dexmedetomidine, a highly selective α-2 adrenergic receptor agonist, acting by inhibiting the release of presynaptic norepinephrine has the therapeutic action of sedation and analgesia with a stability of hemodynamics and mild respiratory depression [19].

Compared with an opioid-only analgesia strategy, the combination of DEX with opioids was beneficial in relieving postoperative pain while reducing the need for opioid use and opioid-related adverse events. Yang confirmed that DEX reduced pain scores for up to 24 h [20]. Adding DEX extended the analgesia effect for up to 48 h in our study which was attributed to multimodal analgesia. More importantly, DEX played a key role in reducing cognitive decline up to one postoperative month in elderly patients undergoing scheduled laparotomy [24]. A meta-analysis investigating the effect of DEX on POCD found that DEX reduced the incidence of POCD and increased postoperative Mini-mental State Examination (MMSE) score [29]. A recent study was conducted to determine that the administration of DEX considerably improved the MMSE and Montreal Cognitive Assessment (MoCA) scores and reduced early POCD (72 h after intubation) [30]. 

The MoCA scale was developed by Nasreddine referring to each subitem of the MMSE scale, and was mostly used for the assessment of mild cognitive impairment [31]. MoCA is more sensitive than MMSE in identifying mild neurocognitive impairment. MoCA-B facilitates screening people with illiteracy and low education. Since patients included in our study were less educated, MoCA-B was especially suitable for evaluating POCD in our research. However, our research did not find that the DEX group showed significantly higher MoCA-B scores than the non-DEX group after discharge, indicating that two days of injection of DEX would improve short-term POCD rather than long-term POCD. It should be noted that this short-term dysfunction can develop into permanent cognitive impairment, such as Alzheimer’s disease (AD), which can lead to the loss of independent living and cause severe physical and psychological damage after surgery [32,33].

Although many pain-relieving remedies have been employed as regular measures, it is still a challenge to optimize postoperative pain control. The use of multimodal analgesia conformed to the enhanced recovery after surgery (ERAS) paradigm that aims to reduce the stress response, decrease postoperative complications, shorten the LOS, and accelerate postoperative rehabilitation [34,35]. Our study demonstrated that LIA plus a single FNB in combination with continuously intravenous infusion of DEX perioperatively prolonged analgesic effect, reduced opioid consumption, lowered VAS pain scores, expedited first time of ambulation after the surgery, javascript:void(0); increased MoCA-B scores and enhanced ROM in TKA patients.

A study conducted by Mika [36] indicated that a single dose of periarticular LIA contributed to improvement of early knee ROM. In addition, the LIA resulted in a significantly reduced LOS. Previous studies [37,38] showed that LIA decreased maximum pain VAS scores and diminished opioid consumption, which in return lowered the incidence of opioid-related side effects and decreased LOS. In our study, the combination of DEX and the anesthetic mixture of LIA alleviates postoperative pain and promotes convalescence; it also shortened LOS for patients undergoing TKA. The earlier patients were able to start rehabilitation training, the less time they stayed in hospital. In contrast to previous studies [37,38], the amount of local anesthetics used for LIA and FNB was lower in our study because we combined it with continuous i.v. injection of dexmedetomidine for pain control to avoid anesthetic intoxication and the multimodal analgesia method that extended pain control to as far as 48 h postoperatively. Sufficient pain alleviation promoted early rehabilitation for sufferers which encouraged patients to exercise active and passive knee joint and advanced recovery of muscle tension. With the early dirigation of knee joint, patients were able to walk with a walker for more than 50 feet and knee range of motion greater than 90 degrees with VAS score less than 5 grades during active ROM exercises (within the range of tolerance). Therefore, DEX group in our research had shortened hospitalization.

PONV is often considered to be the most common adverse reaction caused by high consumption of narcotics. Diiorio [39] observed that the incidence rate of PONV was 39–70% in orthopedics. Although we now pay more attention to postoperative analgesia, recent studies [40,41] indicated that 13–60% of patients still suffered from moderate to severe postoperative pain. To reduce PONV as well as to increase postoperative pain control, we added periarticular LIA to FNB for patients undergoing TKA. Adequate pain relief and less narcotics consumption gave rise to less PONV. Not only did DEX, continuously infused from the beginning of induction to POD 2, improve analgesia, but it would reduce the incidence of PONV [42]. In particular, there was a significant decrease in narcotics consumption (for rescue analgesia) in group A compared to group B on PODs 1 and 2 (when the most severe pain occurred). More patients in group B experienced PONV than those in group A (3 vs. 11, χ^2^ = 5.166, *p* = 0.023).

Adequate pain relief and fewer complications accelerated the start of functional exercise. Surgical outcome, in terms of postoperative ROM, was also greater in DEX group than the non-DEX group (group B), which could have been an indirect result of better analgesia. Equally, better pain relief in group A resulted in better ROM than that of group B.

Few studies have explored the HSS scores following TKA with FNB and LIA in combination with DEX. In our study, group A had higher HSS scores than group B at 1 month and 1 year after the operation, which indicated accelerated rehabilitation after adding DEX to appropriate anesthetic technique (i.e., LIA and FNB).

This study had several limitations. The working concentration and optimal volume of local anesthetics could be obtained if plasma concentrations of infiltration drugs were measured, which was not done. In addition, we did not test inflammatory factors to investigate the effects of DEX on postoperative POCD. To make it a better multimodal analgesia for TKA patients, future studies are therefore required to determine 1. the optimal concentration and volume of LIA and 2. which inflammatory factor and brain-derived neurotrophic factor have therapeutical effect on POCD.

## 5. Conclusions

Combining local infiltration analgesia and femoral nerve block with intravenous DEX for TKA patients relieves postoperative pain, facilitates early rehabilitation, increases ROM, decreases PONV, shortens the LOS and reduces POCD.

## Figures and Tables

**Figure 1 brainsci-12-01652-f001:**
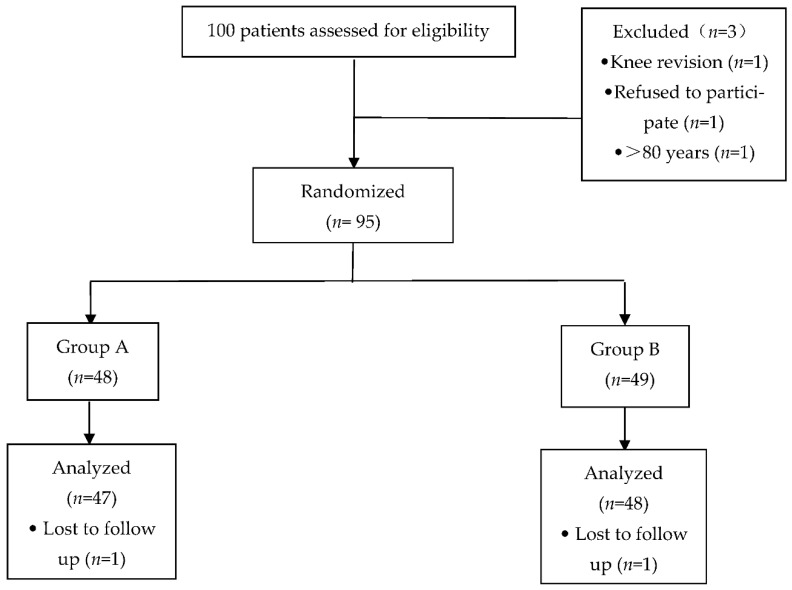
Patient group assignment and result. The reasons of withdrawal are indicated.

**Table 1 brainsci-12-01652-t001:** Basic demographics and medical information of patients.

	Group A (*n* = 47)	Group B (*n* = 48)	*p* Value
Age (yr)	68.2 ± 6.0	67.2 ± 5.2	0.373
Gender (M/F)	6/41	9/39	0.426
Height (cm)	160.0 ± 4.6	160.5 ± 6.2	0.645
Weight (kg)	63.9 ± 7.0	62.8 ± 5.3	0.389
BMI (kg/m^2^)	24.9 ± 2.2	24.3 ± 1.4	0.165
ASA (I/II/III)	7/22/18	9/20/19	0.907
Remifentanil consumption (mg)	0.75 ± 0.13	0.78 ± 0.12	0.287
Duration of surgery (min)	75.7 ± 6.7	76.4 ± 7.4	0.615

Data are expressed as means ± SDs or numbers of patients as appropriate. BMI, body mass index; ASA, American Society of Anesthesiologists.

**Table 2 brainsci-12-01652-t002:** Mean VAS scores at rest for each group at different time points.

	Group A (*n* = 47)	Group B (*n* = 48)	*p* Value
PRE	1.36 ± 0.76	1.44 ± 0.77	0.63
POD1	1.43 ± 0.77	1.60 ± 0.64	0.44
POD2	1.49 ± 0.78	1.81 ± 0.82	0.02
DIS	1.60 ± 0.71	1.81 ± 0.61	0.11
POM1	1.28 ± 0.68	1.54 ± 0.68	0.06
POY1	0.94 ± 0.73	1.04 ± 0.74	0.49

Data are expressed as means ± SDs. BMI, body mass index; ASA, American Society of Anesthesiologists.

**Table 3 brainsci-12-01652-t003:** Mean VAS scores with movement for each group at different time points.

	Group A (*n* = 47)	Group B (*n* = 48)	*p* Value
PRE	5.23 ± 0.70	5.27 ± 0.71	0.80
POD1	3.15 ± 0.59	4.56 ± 0.90	0.00
POD2	3.62 ± 0.64	4.81 ± 0.89	0.00
DIS	3.53 ± 0.88	4.13 ± 0.70	0.00
POM1	3.43 ± 0.85	3.69 ± 0.83	0.13
POY1	3.32 ± 0.91	3.65 ± 0.91	0.08

Data are expressed as means ± SDs. PRE: preoperative period, POD: postoperative day, DIS: discharge, POM1: 1 month postoperatively, POY1: 1 year postoperatively.

**Table 4 brainsci-12-01652-t004:** Secondary Outcomes.

	Group A (*n* = 47)	Group B (*n* = 48)	*p* Value
Narcotics consumption/mg			
POD 1	9.4 ± 2.5	16.8 ± 4.4	0.000
POD 2	17.9 ± 3.8	38.8 ± 4.7	0.000
First time of ambulation/day	1.1 ± 0.63	1.6 ± 0.67	0.000
PONV	3	11	0.023
LOS/days	4.2 ± 0.7	5.0 ± 0.8	0.000

Values are expressed as mean ± SDs. POD: postoperative day, PONV: postoperative nausea and vomiting, LOS: length of hospital stay.

**Table 5 brainsci-12-01652-t005:** The knee range of motion (°) at different periods for each group.

	Group A (*n* = 47)	Group B (*n* = 48)	*p* Value
PRE	97.26 ± 13.37	98.73 ± 14.59	0.609
POD1	108.51 ± 9.92	102.15 ± 10.40	0.003
POD2	114.47 ± 8.80	104.77 ± 10.18	0.000
DIS	123.40 ± 6.44	110.31 ± 7.75	0.000
POM1	121.60 ± 5.62	109.06 ± 9.43	0.000
POY1	119.81 ± 6.23	107.79 ± 8.96	0.000

Data are expressed as means ± SDs. PRE: preoperative period, POD: postoperative day, DIS: discharge, POM1: 1 month postoperatively, POY1: 1 year postoperatively.

**Table 6 brainsci-12-01652-t006:** MoCA-B scores at each time point between two groups.

	Group A (*n* = 47)	Group B (*n* = 48)	*p* Value
PRE	27.17 ± 1.42	27.30 ± 1.02	0.677
POD1	23.57 ± 1.25	22.50 ± 2.00	0.016
POD2	24.77 ± 1.14	23.50 ± 1.89	0.003
DIS	25.37 ± 1.03	24.60 ± 1.63	0.034
POM1	26.73 ± 1.05	26.70 ± 1.32	0.914
POY1	27.07 ± 0.94	27.17 ± 1.12	0.709

Data are expressed as means ± SDs. PRE: preoperative period, POD: postoperative day, DIS: discharge, POM1: 1 month postoperatively, POY1: 1 year postoperatively.

**Table 7 brainsci-12-01652-t007:** HSS scores at different time points.

	Group A (*n* = 47)	Group B (*n* = 48)	*p* Value
PRE	69.13 ± 2.97	68.17 ± 3.46	0.150
POM1	78.40 ± 3.00	74.52 ± 3.41	0.000
POY1	76.77 ± 4.40	72.69 ± 3.37	0.000

Data are expressed as means ± SDs. PRE: preoperative period, POM1: 1 month postoperatively, POY1: 1 year postoperatively.

## Data Availability

The data presented in this study are available on request from the corresponding author.

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
