# Peer review of "Multimodal Analgesia for Accelerated Rehabilitation after Total Knee Arthroplasty: A Randomized, Double-Blind, Controlled Trial on the Effect of the Co-Application of Local Infiltration Analgesia and Femoral Nerve Block Combined with Dexmedetomidine"

_brainsci, 2022, doi:10.3390/brainsci12121652_

Round 1

Reviewer 1 Report

The authors have developed an interesting double-blind RCCT that aimed to investigate the effect of local infiltration analgesia plus femoral nerve block and co-application of perioperative dexmedetomidine on total knee arthroplasty outcomes.

However, I suggest several clarifications and modifications that will in my opinion improve the quality of their manuscript.

1.It is not mandatory, but I think the title would be better shorter.

2. When you mention treatment by arthroplasty in the introduction, I recommend the authors to comment before conservative treatment, which can be very useful in mild-moderate OA. I recommend using and mentioning the following quality papers of studies investigating the use of Therapeutic Exercise and Pain Education: DOI: 10.3390/ijerph19106194, DOI: 10.3390/jcm10112229

3. I recommend the authors to start the Discussion by commenting the main results of their study.

4. It would be possible to add a small "Future Directions" section at the end of the "Discussion". 

5. Could you separate the "Conclusions" section into important points? In addition, please state the conclusions of the study only.

Author Response

Dear reviewer,

We would like to thank you for your careful and thorough reading of this manuscript and for the thoughtful comments and constructive suggestions, which help to improve the quality of this manuscript. In this revised version, revisions to the manuscript had been marked up by *“Track Changes”* function. Our response are as follows:

Reviewer 2 Report

Multimodal analgesia for accelerated rehabilitation after total knee arthroplasty: A randomized, double-blind, controlled trial on the effect of the co-application of local infiltration analgesia and femoral nerve block combined with dexmedetomidine

Multimodal postoperative pain regimens are widely used following total knee arthroplasty (TKA). However, there are few studies on the rehabilitation of the co-application of local infiltration analgesia (LIA) and femoral nerve block (FNB) combined with dexmedetomidine (DEX) for patients undergoing TKA.

The authors aimed to investigate the effects on rehabilitation of administering a combination of local infiltration analgesia (LIA) and femoral nerve block (FNB), with (47 patients) and without (48 patients) dexmedetomidine (DEX)(given by continuous intravenous injection starting from the induction of anaesthesia to postoperative day 2), in a randomized, double-blind, controlled trial of patients undergoing total knee arthroplasty.

All patients were allowed patient-controlled analgesia postoperatively. Those patients who received DEX had lower mean Visual analog scale (VAS) scores at rest and with movement, a better knee range of motion (ROM) degrees at every time point (which continued beyond one year post-op), a reduced narcotic consumption, reduced length of hospital stay (LOS), better Hospital for Special Surgery (HSS) scores and increased Montreal Cognitive Assessment-Basic (MoCA-B) Scores(all P<0.05). Conclu- 28 sion:Our study indicated multimodal analgesia involving a single FNB and LIA combined with 29 DEX accelerates rehabilitation for patients undergoing TKA><0.05)

The authors concluded that their multimodal analgesia involving a single FNB and LIA combined with DEX accelerated rehabilitation for patients undergoing TKA

This is a very interesting paper as we are always keen to improve patient outcomes and this combination appears to be a significant change for the better.

I have some comments:

1)

In the era when patients are being discharged on PO Day zero and PO Day 1, where do the authors see DEX having a role? Quite clearly PCA’s and IV infusion are therefore not possible.

I am not sure that the authors can generalise that using DEX reduced hospital expenditure for all. Line 325. They should clarify that it would reduce expenditure in their own facility. 4.2 days LOS v 5 days in their particular establishment.

2)

There are some spelling mistakes and the whole paper should be scanned.

For example in the Abstract “form the induction” should be “from the induction”.

Same mistake on Line 89.

3)

Lines 56-57

In the Introduction can the authors briefly explain what dexmedetomidine is, and how it is thought to work?

4)

Lines 85 and elsewhere in the text.

Why were the groups named D and F? This is not a standard nomenclature.  One might instead use A and B, or 1 and 2.

5)

Lines 89-91

How were the patients given a continuous IV infusion. Was this only through the PCA pump, in which case the administration would not have been necessarily steady/continuous.

6)

Lines 89.

A knee replacement operation takes around 60mins. Can the authors clarify how the anaesthetist judged when to stop the continuous IV injection? Was this, for example, when the components started being cemented? This is usually about 20-30 mins before the operation ceases.

7)

In effect most patients would only receive one initial bolus and then only a small amount of intraoperative addition DEX.

Did the authors consider giving just the initial induction bolus dose, rather than the small continuous intraoperative infusion. What was the point of having the intraoperative DEX?

8)

There would have been a gap in DEX administration between the induction, the intraoperative and then the start of PCA DEX. When was the PCA commenced? Was this gap in administration important? Can the authors comment?

9)

It would be useful to know how much PCA volume was actually used in each patient? Did any require a “top up” from the base infusion rate?

Patient might have received: 2mls per hour (24) x 2 days = 96 mls in total as a base infusion.

Maximum potential boluses could have been 2ml every 15mins (8mls per hours x 48 hours or a total of 384mls!).

10)

Can the authors clarify the actual total duration that patients received DEX? This is not entirely clear in the manuscript. Did this vary per patient? It is licensed for used for less than 24 hours currently.

 11)

Were there any negatives to using DEX?

Author Response

Dear reviewer,

Thank you for your positive and constructive comments and suggestions on our manuscript entitled “Multimodal analgesia for accelerated rehabilitation after total knee arthroplasty: A randomized, double-blind, controlled trial on the effect of the co-application of local infiltration analgesia and femoral nerve block combined with dexmedetomidine”. We have carefully revised the manuscript according to your comments. Based on the suggestions, we have made an extensive modification on the revised manuscript. The changes to our manuscript were marked up by *“Track Changes”* function. Detailed revision was shown as follows.

Round 2

Reviewer 1 Report

The authors have substantially improved their manuscript over the previous version, so I recommend their work for publication.

Congratulations